# Data-driven Multi-Fidelity Modelling for Time-dependent Partial Differential Equations using Convolutional Neural Networks

**Freja T. Petersen and Allan P. Engsig-Karup**
DTU Compute
Technical University of Denmark (DTU)
Kgs. Lyngby, DK

## Abstract

We present a general multi-fidelity (MF) framework which is applied through utilizing flexible-order explicit finite difference numerical schemes using convolutional neural networks (CNNs) by combining low-order simulation data with higher order simulation data obtained from numerical simulations based on partial differential equations (PDEs). This allows for improving the performance of low-order numerical simulation through learning from the data how to correct the numerical schemes to achieve improved accuracy. Through the lens of numerical analysis we evaluate the accuracy, efficiency and generalizability of constructed data-driven MF-models. To illustrate the concept, the construction of the MF models uses CNNs and is evaluated against numerical schemes designed for solving linear PDEs; the heat, the linear advection equation and linearized 1D shallow water equations. The numerical schemes allow for a high level of explainability of data-driven correction terms obtained via CNNs through numerical analysis of truncation errors. It is demonstrated that data-driven MF models is a means to improve the accuracy of LF models through operator correction.

## 1 Introduction

Convolutional neural networks (CNNs) are kernel-based local approximators which can be applied in the context of numerical solvers for partial differential equations. We investigate the usability of CNNs in a multi-fidelity (MF) setting using explicit flexible-order finite difference numerical solvers. The aim of generating a multi-fidelity model is to take advantage of combining expensive and often sparse high-accuracy data with cheap and easily obtainable low-accuracy data to improve accuracy of lower fidelity numerical models.

### 1.1 Related work

Decades ago, neural networks were established as function approximators by Hornik et al. (1989), which was later established in practice by Rico-Martinez et al. (1995) and González-García et al. (1998). This is fundamental for solving e.g. dynamical systems and differential equations using neural networks. More recently, the universality of CNNs was established (Zhou, 2020).

Recently, the use of multi-fidelity techniques has been evolving using neural network based approaches. The term *multi-fidelity techniques* refers to methods that operate with different fidelities. The word *fidelity* is a broad term, which describes the overall quality or "truthness" of the model. The higher the fidelity, the closer to the truth the model is. The fidelity of the model can be controlled in terms of grid-discretization, order of the numerical method, and how well a model approximates the true system. Examples of previous neural network-based multi-fidelity models include a composite neural network that learns from multi-fidelity data (Meng & Karniadakis, 2020), a long short-term memory network for multi-fidelity surrogate modeling (Conti et al., 2023) and a neural network-based acceleration of a high-order discontinuous Galerkin solver (de Lara & Ferrer, 2023). In the context of initial value problems for PDEs, methods using multi-resolution wavelet operator learning

have been successful in beating state-of-the-art in terms of obtaining high accuracy for flexible resolutions (Gupta et al., 2021; 2022; Xiao et al., 2023). All of these methods use multi-layer-perceptron (MLP) type networks, which are global in nature and hence difficult to scale.

CNNs, on the other hand, use localized kernels that are executed across the domain. The advantage of this is that the computational complexity scales linearly with the input dimension. There are a few examples of CNN-based MF-models, including the a deep CNN-based MF-model for temperature field prediction (Zhang et al., 2023) and a CNN-structure called Deep FDM for enhancing finite difference methods (Kossaczká et al., 2023). The latter argues for a strong connection between CNNs and numerical schemes. This work builds upon this idea, and explores it even further.

## 1.2 CONTRIBUTIONS

The above section describes the use of neural networks and multi-fidelity models for solving PDEs. This work contributes to the area by creating a strong connection between numerical theory and CNNs, and the contributions include: 1) the formulation of a general framework for developing multi-fidelity models for future research and 2) data-driven multi-fidelity experiments with numerical simulators of different complexities. This work serves as a proof of concept, highlighting the potential of the proposed CNN-based multi-fidelity methods.

## 2 METHODOLOGY

### 2.1 A GENERAL MULTI-FIDELITY FRAMEWORK

Consider the general initial value problem (IVP):

$$\frac{\mathrm{d}\boldsymbol{u}}{\mathrm{d}t} = \boldsymbol{f}(\boldsymbol{u}(t), t), \quad \boldsymbol{u}(t_0) = \boldsymbol{u}_0, \quad t > 0, \tag{1}$$

where $\boldsymbol{u} \in \mathbb{R}^n$ holds $n$ state variables and $\boldsymbol{u}_0 = \boldsymbol{u}(t_0)$ is the initial state at time $t = t_0$, and $f(\cdot)$ is assumed Lipschitz continuous with respect to the first argument. In the setting of time-dependent partial differential equations, it is common to discretize the PDEs using the method of lines (MoL) in two steps, where the first step involves numerical discretization in space to generate a semi-discrete system in the form of an IVP. The subsequent step, is to then solve this spatially discretised system via finite differences in the form of ordinary differential equation (ODE) solvers that provide stability and efficiency for the temporal integration. The general solution to the IVP can be stated as

$$\boldsymbol{u}(t_n) = \boldsymbol{u}(t_0) + \int_{t_0}^{t_n} \boldsymbol{f}(\boldsymbol{u}(t), t)dt. \tag{2}$$

For governing equations such as nonlinear PDEs, the integral cannot be readily evaluated. Instead it is possible to resort to numerical approximation.

Following Chen & Xiu (2021) that introduced the generalised Residual Neural Networks (gResNet) for data-driven modeling of autonomous dynamical systems, we introduce an operator, $\mathcal{L} : \mathbb{R}^n \to \mathbb{R}^n$, such that:

$$\boldsymbol{u}(t_{k+1}) \approx \mathcal{L}(\boldsymbol{u}(t_k)) \quad \leftrightarrow \quad \boldsymbol{u}(t_{k+1}) = \mathcal{L}(\boldsymbol{u}(t_k)) + \tau,$$

where $\tau$ is the error (TE) of the numerical scheme that is dependent on the underlying solution $\boldsymbol{u}$. To perform numerical analysis, the TE can be expanded in terms of a Taylor series to understand the structure of the leading order terms wrt. to the numerical discretization parameters.

In the context of MF-models, $\mathcal{L}$ can be considered a low-fidelity (LF) operator, which maps the state variables in a time step from time $t_k$ to time $t_{k+1} = t_k + \Delta t$ with $\Delta t$ the time step. The LF operator is assumed to be a *convergent numerical scheme* for finding approximate solutions to the IVP equation 1. Consider now the operator: $\mathcal{F} : \mathbb{R}^n \to \mathbb{R}^n$. This operator can be subject to function approximation, e.g., via numerical discretization or neural networks, and the update takes the form:

$$\boldsymbol{u}(t_{k+1}) = \mathcal{L}(\boldsymbol{u}(t_k)) + \mathcal{F}(\boldsymbol{u}(t_k)). \tag{3}$$

In the framework of equation 3, $\mathcal{F}(\boldsymbol{u}(t_k)) \approx \tau$ can be interpreted as a correction to the LF operator $\mathcal{L}(\boldsymbol{u}(t_k))$, and the framework is therefore characterized as a multi-fidelity model when the correction is constructed based on utilizing data of different fidelities in a learning procedure.

## 2.2 A CNN-BASED MULTI-FIDELITY FRAMEWORK

Consider a method of lines discretization of the IVP in equation 1 onto a uniform grid, such that $u(x_j, t_i)$ denotes the $j$th grid point of the spatial discretization and $i$ is the time step. Consider an explicit low-order FDM with a stencil of half-width $K$ and a TE, $\tau_{FDM}$. A temporal step with the FDM approximating equation 2 is:

$$u(x_j, t_{i+1}) = F_{FDM}(u(\bar{x}_j, t_i)) + \tau_{FDM},$$

for $i = 0, 1, ..., N_T$ and $j = 0, ..., m+1$ and where $\bar{x}_j = [x_{j-K}, ..., x_j, ..., x_{j+K}]$. The CNN-based MF-model consists of the FDM being the LF-operator, $\mathcal{L}$, and an additive correction term represented via a CNN being an approximation to the TE term $\tau_{FDM}$. For example, the TE of the low-order FTCS-scheme for the heat equation is $(\frac{\Delta t}{2} - \frac{\Delta x^2}{12})\frac{\partial^4 u}{\partial x^4}$. This term is the difference between a fourth order and a second order method, which use the same time-stepping scheme. It is possible to approximate this TE using a local finite difference stencil for the fourth derivative and apply this stencil anywhere on the domain or for any initial value. The motivation behind using a CNN for approximating this term is that the convolutional layers consist of local kernels similar to those of a finite difference stencil, except they are trained using data from the higher-order numerical method. This work therefore investigates to what extent a CNN can represent the truncation error of the low-order method compared to a high-order method, which uses a different time-stepping scheme, and hence obtain a kernel-based local approximator to the truncation error.

The architectures of the CNNs used for the testing in the following consist only of convolutional layers and activation functions (as described in appendix A.2). To limit the computational expense for the *online* phase of the simulation, the CNNs should be relatively shallow (i.e. few layers). Ultimately, a CNN of only two layers with five channels, kernels of size five and no activation functions is selected.

## 2.3 TEST EQUATIONS: TIME-DEPENDENT LINEAR PARTIAL DIFFERENTIAL EQUATIONS

The MF-framework is tested on the one-dimensional heat equation and the one-dimensional linear advection equation as well as the system of equations given by the linearized one-dimensional shallow water equations (LSWE) assuming small-amplitude waves. The heat equation is given by:

$$\frac{\partial u(x,t)}{\partial t} = \kappa \frac{\partial^2 u(x,t)}{\partial x^2}, \quad u(x,0) = u_0, \tag{4}$$

where $\kappa$ is the diffusion coefficient. The advection equation is given by:

$$\frac{\partial u(x,t)}{\partial t} = -v \frac{\partial u(x,t)}{\partial x}, \quad u(x,0) = u_0, \tag{5}$$

where $v$ is the advection velocity. The LSWE are given by:

$$\frac{\partial u}{\partial t} = -g \frac{\partial \eta}{\partial x}, \quad \frac{\partial \eta}{\partial t} = -d \frac{\partial u}{\partial x} \tag{6}$$

where $g$ is the local graviational acceleration and $d$ is the mean depth of the water and with initial values $u(x,0) = u_0$ and $\eta(x,0) = \eta_0$. $u$ is the velocity and $\eta$ is the surface elevation of the water.

Implementation details are found in appendix A.3. Table 1 shows the analytical solutions for testing the efficiency and accuracy of the MF-models for the heat and advection equation. The solutions are periodic and described on the domain $x \in [-\pi, \pi]$. The numerical models are initialized using the analytical solutions at time step 0, $u_1(x,0)$ and $u_2(x,0)$, respectively. The numerical solution corresponding to $u_1$, is used for *training* the MF-model, whereas $u_2$ is only used for testing the *generalization* of the MF-model to unseen solutions. The data for training and testing is generated by sampling values of $a$, $b$ and $c$ from a uniform distribution. For the LSWE, a traveling wave solution is used as described in table 2 on the domain $x \in [0, 1]$. The constants are described in table 5 in appendix, where $H$ is sampled in order to introduce variation in the training data.

The equations are simulated using explicit finite difference methods (FDMs). The CNN is trained to correct a second or first order FDM using a fourth order FDM as reference. Hence, this paper relies on the use of flexible order FDMs and the strength of the MF-model is to avoid the strict stability requirements of using a higher order method. In the case of the LSWE, the HF-data used for training is based on a finer spatial and temporal discretization step by doubling the number of spatial grid points, resulting in a MF-model which captures both a higher order accuracy as well as a finer discretization. The training procedure is presented in appendix A.4

Table 1: Analytic, periodic solutions to the heat and advection equation.

|  | $u_1(x,t)$ | $u_2(x,t)$ |
|---|---|---|
| Heat | $c + ae^{-b^2\pi^2 t}\sin(b\pi x)$ | $c + a\cos(b\pi x)e^{-b^2\pi^2 t} + \sin(\frac{b}{4}\pi x)e^{-\frac{b^2}{16}\pi^2 t}$ |
| Advection | $a\sin(b\pi(x-vt))$ | $a\cos(b\pi(x-t)) + \sin(\frac{b}{4}\pi(x-t))$ |

Table 2: Analytic, traveling wave solution to the LSWE.

|  | $u(x,t)$ | $\eta(x,t)$ |
|---|---|---|
| LSWE | $\frac{\pi HL}{2\pi Th}\cos(\omega t - kx)$ | $\frac{H}{2}\cos(\omega t - kx)$ |

## 3 RESULTS AND DISCUSSION

An MF-model is trained on data for the initial value function $u_1(x,0)$ for different numbers of interior grid points, $m$, for the two equations. Followed by this, the MF-model is tested by simulation with 100 initial values with new samples of $a$, $b$, and $c$ up to a time $T$ for both $u_1$ and $u_2$, where the purpose of the latter is to assess the generalization to other functions. In the top row of figure 1, the results for $u_1$ are seen, and in the bottom row the results for $u_2$ are seen. The figures show the the infinity norm of the mean absolute errors of the LF, MF, and HF-model. The data behind the figure is found in appendix A.5. For the advection equation, a second order Lax-Wendroff method is also tested (LF (2)) for comparison. The bars show the number of useful floating point operations (FLOPs) needed for the time-stepping of each of the models.

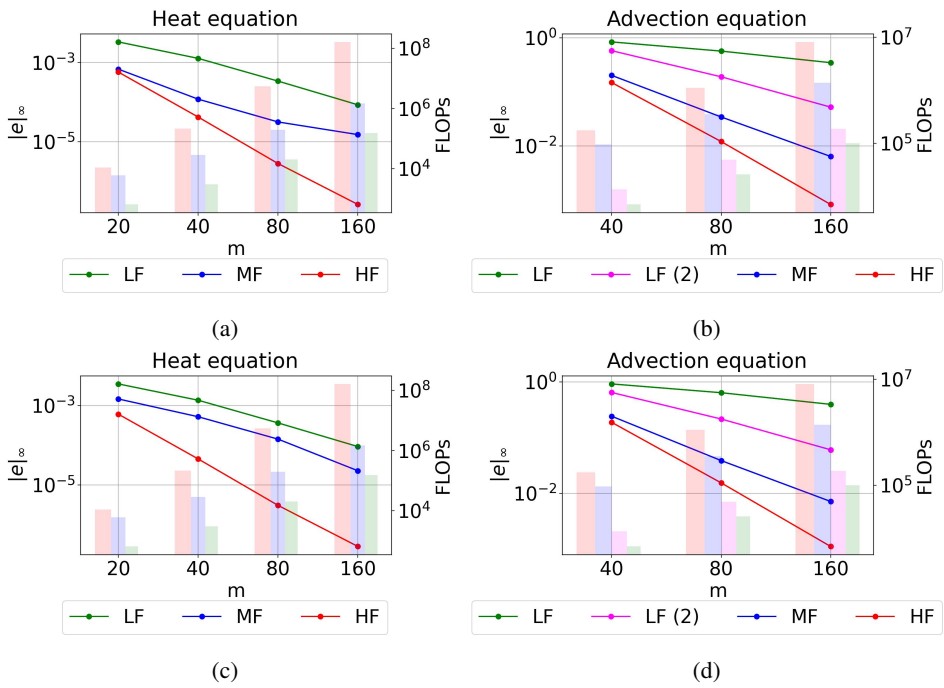

Figure 1: Infinity norm of the absolute errors of the three models in a double-logarithmic plot for the heat and advection equation with the initial value function $u_1(x,0)$ on the top (a and b) and the initial values $u_2(x,0)$ on the bottom (c and d). The bars show the number of FLOPs (right $y$-axis). The numbers are mean values across 100 (a, b) and 20 (c, d) simulations.

Considering the top row, the MF-model achieves better performance both in terms of accuracy and efficiency. The MF-model improves the error of the LF-model, while using fewer FLOPs than the HF-model, since it avoids the strict stability requirement. For the heat equation, there seems to be a saturation point in terms of accuracy, since the slope of the graph decreases with $m$. For the

advection equation, it seems that a second order convergence rate is obtained, yet with lower error than the second order Lax-Wendroff scheme. These results show that the MF-model serves as an improved numerical scheme with properties (truncation errors and computational requirements) that are midway between known numerical schemes.

The bottom row, showing results for second initial value function $u_2(x, 0)$ without retraining the CNN, is promising. It is seen that the error of the MF-model is still lower than the error of the LF-model. Though, for the heat equation, the errors are closer to the LF-model than in figure 1a. For the advection equation, the result is approximately the same as in figure 1b, despite the fact that the CNN is trained on a completely different initial value problem. This indicates a property of generalizability of the CNN to unseen initial values. Considering the trade-off between FLOPs and errors, the advantage of the MF-model is not as clear for the heat equation as it is for the advection equation. For the heat equation, the computational requirement of the MF-model is midway between the LF- and HF-model, but the HF-model achieves much better accuracy. Hence, the MF-model would probably not be the first choice. Though, for the advection equation, the generalization properties together with the fact that the error of the MF-model is comparable to the HF-model while using fewer FLOPs, the advantage of the MF-model is more clear.

Figure 2 shows the result for the LSWE. The plot shows that the CNN is able to accurately approximate the TE of the high-order model with double the amount of spatial grid points ($m$), resulting in fourth order accuracy for the MF-model and a lower computational effort compared to the HF-model. In the optimized framework of PyTorch (software information is found in app. A.1), a simulation with the MF-model with $m = 20$ takes 0.11 seconds (mean value for 100 simulations) and a simulation with the HF-model with $m = 40$ takes 0.29 seconds. Thus, the same accuracy can be achieved with a speed-up of a factor 2.6 using the MF-model. It is possible that further speed-ups could be achieved by training the CNN on HF-data which uses an even finer discretization.

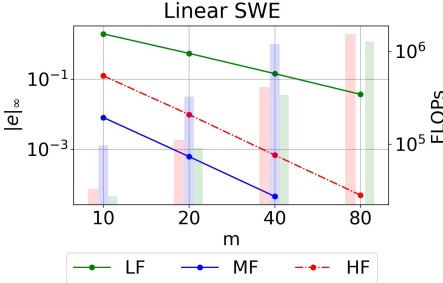

Figure 2: Infinity norm of the absolute errors of the three models for the LSWE in a double-logarithmic plot. The bars show the number of FLOPs (right y-axis).

## 4  CONCLUSION AND OUTLOOK

In conclusion, this paper shows the potential of CNN-based MF-models for achieving an explainable and interpretable MF-model, which succeeds in correcting the error of a low-order explicit finite difference solver to achieve higher order for a set of linear time-dependent PDEs. The paper further shows the potential to combine the use of flexible order methods with varying grid discretizations in order to ensure speed-up of the MF-model. In ongoing work, we will consider the construction of MF-models for nonlinear and more complex time-dependent partial differential equations where flexible-order numerical schemes are utilized for data generation.

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

## A  APPENDIX

### A.1  MACHINE AND SOFTWARE

All implementations of this work use PyTorch (PyTorch, 2023). PyTorch version 2.0.1 is installed with python version 3.10.12. The default precision of PyTorch is single-precision floating-point format (float32). This has been changed to double-precision floating point-format (float64) throughout this work to reduce the influence of round-off errors. The machine used for training and collection of all results is a 2021 Macbook Pro with an Apple M1 Pro chip, with 16 GB RAM memory and with the Mac Ventura operating system version 13.2.1.

### A.2  CNN-STRUCTURE

Figure 3 shows the structure of the CNN used for obtaining the results of this paper. Other structures were tested as well, including CNNs with 2 or 4 convolutional layers with kernels of size $K = 3$ or $K = 5$, both with and without activation functions. The structure in the figure was the most efficient in terms of training time (between 15 minutes and 1 hour in all cases) and had the lowest test error.

For the advection equation, the CNN takes two time steps as input ($N_{in} = 2$) and in the other cases, only the previous time step is taken as input ($N_{in} = 1$). The CNN takes the current state as input and outputs a correction to the FDM approximation of the next time step.

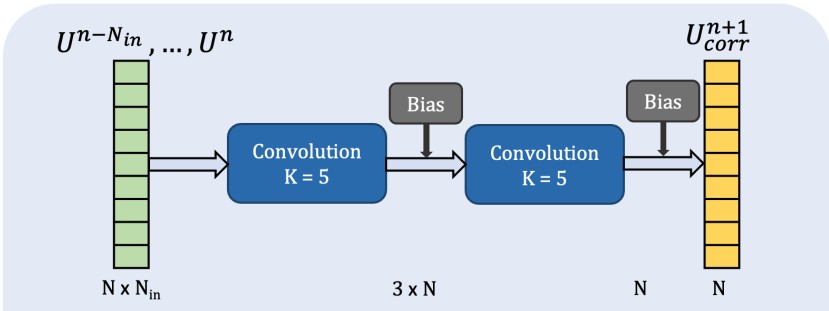

Figure 3: Diagram of the CNN-structure used for obtaining the results of this section

### A.3  IMPLEMENTATION DETAILS

The heat equation is simulated using a forward-time-central-space (FTCS) discretization, which is first order accurate in time, and second and fourth order accurate in space for the low- and high-fidelity simulations, respectively. The low-order simulation for the advection equation is based on the upwind method, which is first order accurate in time and space, and the high-fidelity simulation is based on the Lax-Wendroff method, which is fourth order accurate in the spatial discretization. The time-step, $\Delta t$, is chosen small enough such that the order of the method corresponds to the spatial order. This means that the fourth order methods use a more fine time-step than the low-order methods. The LSWE are simulated using an explicit fourth order Runge-Kutta method for the time integration and a second and fourth order scheme for the spatial discretization of the low and high-fidelity models, respectively. The time-stepping of the LF- and HF-models is the same.

Tables 3 and 4 show specifications of the simulation of the heat and advection equation. The ranges of $a$, $b$, and $c$ used for simulation of the two equations are: For the heat equation, $a \in [1, 2], b \in [0.3, 0.5]$, and $c \in [0, 0.25]$, and for the advection equation, $a \in [0.5, 2]$ and $b \in [0.5, 2.5]$. In both cases, the grid spacing, $\Delta x$, is determined by the x range and the number of interior grid points, $m$. The time-step is determined using the relation $\Delta t = c(\Delta x)^q$, where $c$ and $q$ are given in the table. Note, even though these equations can be used to model physical systems, this work is primarily an academic contribution, and, therefore, units are left out. In a physical system the range of the variable $x$ would be given in meters and the time would be given in seconds.

Table 3: Specifications of the simulation domain for the two test-equations. $N_{train}$ is the size of the training set, $N_{val}$ is the size of the validation set, $T_{end}$ is the end-time of the simulation, x-range is the range of the spatial domain.

| Equation | $N_{train}$ | $N_{val}$ | $T_{end}$ | x range |
|---|---|---|---|---|
| Heat | 100 | 50 | 0.1 | $[-\pi, \pi]$ |
| Advection | 100 | 50 | 1 | $[-\pi, \pi]$ |

Figure 4 shows a comparison of the two initial value functions used for training the MF-model ($u_1$) and testing the generalization ($u_2$), respectively.

Table 5 shows the characteristic constants of the traveling wave solution to the LSWE.

Table 4: Specifications of the simulation domain for the two test-equations and for each type of model (HF is high-fidelity and LF is low-fidelity). The problem-dependent, $c$, and the order of the numerical scheme, $q$, correspond to the relation $\Delta t = c(\Delta x)^q$, which differ for each equation and model.

| Equation | Model | order | c | q |
|---|---|---|---|---|
| Heat | LF | 2 | 0.49 | 2 |
| | HF | 4 | 0.49 | 4 |
| Advection | LF | 1 | 0.3 | 1 |
| | HF | 4 | 0.3 | 2 |

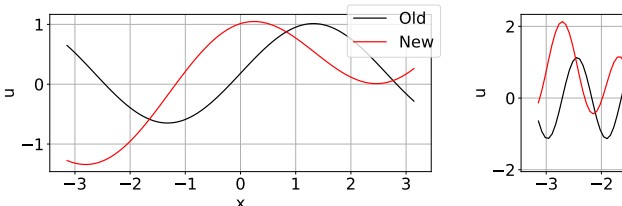

Heat equation, $a = 1.68$, $b = 0.38$, $c = 0.18$.  Advection equation, $a = 1.13$, $b = 1.87$.

Figure 4: Comparison of the initial value functions given by $u_1$ (old) and $u_2$ (new) as given in table 1.

Table 5: Characteristic constants for the wave modeled by the LSWE equation 6.

| Parameter | Description | Value | Unit |
|---|---|---|---|
| $g$ | Local gravitational acceleration | 9.81 | $m/s^2$ |
| $d$ | Mean water depth | 0.25 | $m$ |
| $c$ | Propagation velocity | $\sqrt{gd} \approx 1.57$ | $m/s$ |
| $a$ | Wave amplitude | $a \in [0.2, 1]$ | $m$ |
| $H$ | Wave height | $2a$ | $m$ |
| $L$ | Wave length | 1 | $m$ |
| $T$ | Wave period | $\frac{L}{c} \approx 0.659$ | $s$ |
| $\omega$ | Wave frequency | $\frac{2\pi}{T} \approx 9.84$ | $s^{-1}$ |
| $k$ | Wave number | $\frac{\omega}{\sqrt{gh}} = \frac{2\pi}{L} = 2\pi$ | $m^{-1}$ |

Specifications of the simulation with LSWE can be seen in table 6. The first table shows the domain range and the grid resolution, and the second table shows specifics of the LF- and HF-models. The LF-model is based on a stencil with a half-width of $\alpha = 1$ (stencil of size 3), and the HF-model has a stencil with half-width $\alpha = 2$ (stencil of size 5). The choice of implementation method means that the restrictions on $\Delta t$ in the relation $\Delta t = c(\Delta x)$ is the same for the two models. Figure 5 shows the analytical solution to the LSWE for a chosen value of $a$.

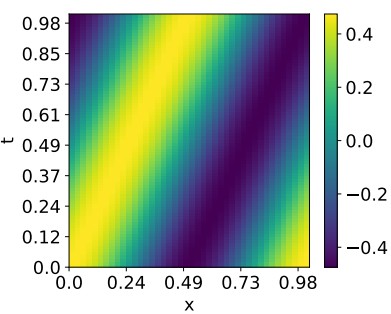 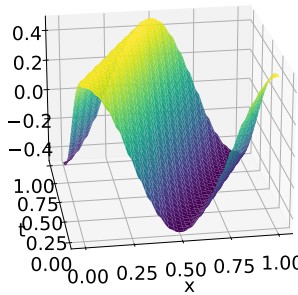

Figure 5: Traveling wave solution to the LSWE. Only the solution for the free surface elevation, $\eta$, is shown.

Table 6: Specifications of the simulation domain the LSWE and for each type of model (HF is high-fidelity and LF is low-fidelity). $N_{train}$ is the size of the training set, $N_{val}$ is the size of the validation set, $T_{end}$ is the end-time of the simulation, x-range is the range of the spatial domain, $m$ is the number of interior grid points and $\Delta x$ is the grid spacing. In the second table, the $\alpha$ column denotes the half-width of the stencil used for the spatial derivative in the explicit fourth-order 5-stage Runge-Kutta method used for the simulation. A higher $\alpha$ means a higher order, since $\alpha$ is the half-width of the finite difference stencil. $\Delta t$ is the time-discretization size, and $c$ corresponds to the relation $\Delta t = c(\Delta x)$.

| Equation | $N_{train}$ | $N_{val}$ | $T_{end}$ | x range |
|---|---|---|---|---|
| LSWE | 100 | 50 | 1 | $[0, 1]$ |

| Equation | Model | $\alpha$ | c |
|---|---|---|---|
| LSWE | LF | 1 | 0.5 |
|  | HF | 2 | 0.5 |

### A.4 TRAINING PROCEDURE

The training procedure is outlined in algorithm 1. One for-loop surrounds two sets of nested for-loops. The outer-most loop, iterates through a number of epochs (iterations), and makes sure that the training and validation procedures are repeated $N_{epochs}$ times. In the first set of nested for-loops, the training procedure is carried out. The outer loop iterates through the time-steps whereas the inner loop iterates through the training data. Thus, the loop is constructed such that *all* training data is visited for one time-step before moving on to the next time-step. For each time step, the function output is calculated using a low-fidelity finite difference method, $F_{LF}$, and an additive CNN, $F_{CNN}$. The loss, when compared to the correct output obtained from the training data, is calculated using a mean-squared-error (MSE) function. The gradient of the loss-function is used to update the weights of the CNN according to the Adam optimizer.

The next set of nested for-loops constitutes the validation procedure. It is only accessed if the training loss is below some tolerance in order to save computation time. For the validation procedure, the outer loop iterates the data and the inner loop iterates the time-steps, such that one set of data is iterated through before moving on to the next. The CNN is not updated in the validation loop. The validation loss is based only on the last time step. If the validation loss is lower than for the previously saved model, the new model is saved.

---

**Algorithm 1** Training procedure for MF-model

---

**Require:** $D_{train}, D_{val}, N_{train}, N_{val}, N_{epochs}, T_{end}, \Delta t, F_{CNN}, F_{LF}, tol, tol_{save}$
    Calculate number of time-steps, $N_T$, using $T_{end}$ and $\Delta t$
    **for** $e$ in range(0, $N_{epochs}$) **do**
        **for** $i$ in range(0, $N_T - 1$) **do**
            **for** $n$ in range(0, $N_{train}$) **do**
                $input \leftarrow D_{train}[n, i]$              ▷ Time-step $i$ in data set $n$
                $U \leftarrow F_{LF}(input) + \Delta t F_{CNN}(input)$     ▷ Corrected time-stepping
                $U_{ref} \leftarrow D_{train}[n, i+1]$     ▷ Reference solution is next time step
                $loss \leftarrow MSE(U_{ref}, U)$          ▷ Calculate loss
                Update weights of $F_{CNN}$ using $\nabla(loss)$
            **end for**
            Calculate $mean(loss)$
        **end for**
        LR_scheduler($mean(loss)$)
        **if** $mean(loss) \leq tol$ **then**         ▷ Condition for performing validation
            $tol \leftarrow mean(loss)$         ▷ Update tolerance level
            **for** $n$ in range(0, $N_{val}$) **do**       ▷ Repeat loops but with validation data
                **for** $i$ in range(0, $N_T - 1$) **do**
                    $input \leftarrow D_{val}[n, i]$
                    $U \leftarrow F_{LF}(input) + \Delta t F_{CNN}(input)$
                    $U_{ref} \leftarrow D_{val}[n, i+1]$
                    $loss_{val} \leftarrow MSE(U_{ref}, U)$
                **end for**
                Calculate $mean(loss_{val})$
            **end for**
            **if** $mean(loss_{val}) \leq tol_{save}$ **then**       ▷ Condition for saving model
                $tol_{save} \leftarrow mean(loss_{val})$
                Save $F_{CNN}$
            **end if**
        **end if**
    **end for**

---

### A.5 RESULTS

Tables 7 and 8 shows the data behind figures 1a, 1b and 2. The FLOPs in table 8 are calculated taking into account only the operations of the FDM and the execution of the CNN (for the MF-model) in each time step, and therefore does not consider other operations such as loading data etc. Table 7 can be used to calculate the convergence rate and thereby determine the order of the methods. This is done by fitting a linear model to the data in the double logarithmic plot, and the slopes of these lines will be an approximation to the order of the model. The results are given in table 9.

Table 7: Mean absolute errors of the models given by the infinity norm across 100 simulations for the heat and advection equations and 20 simulations for the LSWE. Reported with 2 significant digits.

|  | order | 10 | 20 | 40 | 80 | 160 |
|---|---|---|---|---|---|---|
| **Heat** | 2 (LF) | - | $3.3 \cdot 10^{-3}$ | $1.3 \cdot 10^{-3}$ | $3.4 \cdot 10^{-4}$ | $8.5 \cdot 10^{-5}$ |
|  | 2+ (MF) | - | $6.8 \cdot 10^{-4}$ | $1.2 \cdot 10^{-4}$ | $3.2 \cdot 10^{-5}$ | $1.7 \cdot 10^{-5}$ |
|  | 4 (HF) | - | $5.7 \cdot 10^{-4}$ | $4.2 \cdot 10^{-5}$ | $2.9 \cdot 10^{-6}$ | $2.6 \cdot 10^{-7}$ |
| **Advection** | 1 (LF) | - | 1.1 | 0.84 | 0.56 | 0.34 |
|  | 2 | - | 1.2 | 0.58 | 0.19 | $5.3 \cdot 10^{-2}$ |
|  | 1+ (MF) | - | - | 0.20 | 0.03 | $6.4 \cdot 10^{-3}$ |
|  | 4 (HF) | - | 0.95 | 0.15 | 0.012 | $8.2 \cdot 10^{-4}$ |
| **LSWE** | 2 (LF) | 1.98 | 0.550 | 0.145 | 0.0371 | - |
|  | 2+ (MF) | 0.00809 | $6.17 \cdot 10^{-4}$ | $4.48 \cdot 10^{-5}$ | - | - |
|  | 4 (HF) | 0.125 | $9.79 \cdot 10^{-3}$ | $6.80 \cdot 10^{-4}$ | $4.88 \cdot 10^{-5}$ | - |

Table 8: Number of FLOPs for one simulation of each of the three models.

| order — m | 10 | 20 | 40 | 80 | 160 |
|---|---|---|---|---|---|
| **Heat** 2 (LF) | - | 656 | 3040 | 20440 | 154440 |
| 2+ (MF) | - | 6024 | 28660 | 195510 | 1488510 |
| 4 (HF) | - | $1.11 \cdot 10^4$ | $2.141 \cdot 10^5$ | $5.48 \cdot 10^6$ | $1.61 \cdot 10^8$ |
| **Advection** 1 (LF) | - | 2132 | 6992 | 25696 | 99528 |
| 2 | - | 4238 | 13478 | 48664 | 186702 |
| 1+ (MF) | - | - | 94898 | 354024 | 1382082 |
| 4 (HF) | - | 34710 | 175536 | 1105104 | 8160426 |
| **LSWE** LF | 27360 | 91960 | 339150 | 1269770. | - |
| MF | 96768 | 325248 | 1199520 | - | - |
| HF | 33120 | 111320 | 410550 | 1537090 | - |

Table 9: The order of accuracy of the three methods are based on finding the slope of the graphs in figures 1 and 2.

| | model | $p$ for $u_1$ | $p$ for $u_2$ |
|---|---|---|---|
| **Heat** | 2 (LF) | 1.8 | 1.8 |
| | 2+ (MF) | 1.9 | 2.1 |
| | 4 (HF) | 3.9 | 3.9 |
| **Advection** | 1 (LF) | 0.7 | 0.6 |
| | 2 | 1.8 | 1.8 |
| | 1+ (MF) | 2.6 | 2.6 |
| | 4 (HF) | 3.9 | 3.8 |
| **LSWE** | 2 (LF) | 2.1 | - |
| | 2+ (MF) | 4.1 | - |
| | 4 (HF) | 4.1 | - |

