# OpenReview forum: "Data-driven Multi-Fidelity Modelling for Time-dependent Partial Differential Equations using Convolutional Neural Networks"
_ICLR.cc/2024/Workshop/AI4DiffEqtnsInSci — AI4DiffEqtnsInSci @ ICLR 2024 Poster_

### Official Review · Reviewer_zac2 · 2024-02-24
**This article presents a data-driven multi-fidelity (MF) modeling approach that achieves better accuracy compared to LP approach and better FLOPS compared to HP approach on heat equation, linear advection equation, and linearized shallow water equations.**

**Rating:** 7
**Confidence:** 5

**Review:**

A) The authors can use more dimensions in the study to compare LF, HF and MF. From Figure 1 and Figure 2, at some times, the error of HF is much smaller than that of MF, but it's FLOPS is not much greater than that of MF. Therefore, the author needs to explain the superiority of MF from more perspectives.
B) The description of solutions in Section 2.3 is a bit confusing. I hope the author can provide further explanations.
C) The discussion of related work is very thin, there have been many efforts to learn from low fidelity complex PDEs and solve or forecast PDEs at other resolutions and fidelities. Here are some recent examples:
- "Non-linear operator approximations for initial value problems." In International Conference on Learning Representations (ICLR). 2022.
- "Coupled Multiwavelet Operator Learning for Coupled Differential Equations." In The Eleventh International Conference on Learning Representations. 2022.
- "Multiwavelet-based operator learning for differential equations." Advances in neural information processing systems 34 (2021): 24048-24062.
Particularly, the initial value problem has been actively considered in the neural operator literature and so it should be accurately discussed.
D) As the author stated, the performance of the model needs to be verified on more PDEs.
E) What are the main factors that affect the multi-fidelity and why the CNNs are the right approach? I think this is an interesting study but need to be put I the general context so that we can see when we should use it in combination with other techniques.

---

### Meta-Review · Area_Chair_N7Cg · 2024-03-03

**Recommendation:** Accept (Poster)

**Metareview:**

The reviewer clearly marks this paper as a clear accept. I encourage the authors to add the suggested references in the camera ready version and vote for acceptance.

---

### Decision · Program_Chairs · 2024-03-03

Accept (Poster)